# Factors Associated with Uptake of Patient Portals at a Federally Qualified Health Care Center

**DOI:** 10.3390/healthcare12151505

**Published:** 2024-07-30

**Authors:** Alicia K. Matthews, Alana D. Steffen, Jennifer Akufo, Larisa Burke, Hilda Diaz, Darcy Dodd, Ashley Hughes, Samantha Madrid, Enuma Onyiapat, Hope Opuada, Jessica Sejo, Brittany Vilona, Barbara J. Williams, Geri Donenberg

**Affiliations:** 1Department of Population Health Nursing, College of Nursing, The University of Illinois Chicago, Chicago, IL 60612, USA; steffena@uic.edu (A.D.S.); jen.akufo16@gmail.com (J.A.); laburke@uic.edu (L.B.); hdiaz7@uic.edu (H.D.); darcydodd@gmail.com (D.D.); smadri4@uic.edu (S.M.); jonyia2@uic.edu (E.O.); hopuad2@uic.edu (H.O.); vilonab@uic.edu (B.V.); 2Department of Biomedical and Health Information Science, The University of Illinois Chicago, Chicago, IL 60612, USA; amhughes@uic.edu; 3College of Medicine, The University of Illinois Chicago, Chicago, IL 60612, USA; jsejo2@uic.edu (J.S.); gerid@uic.edu (G.D.); 4Cancer Center, The University of Illinois Hospital, Chicago, IL 60612, USA; bwillia2@uic.edu

**Keywords:** patient portals, activation rates, federally qualified health centers, cancer

## Abstract

Federally qualified health centers (FQHC) aim to improve cancer prevention by providing screening options and efforts to prevent harmful behavior. Patient portals are increasingly being used to deliver health promotion initiatives. However, little is known about patient portal activation rates in FQHC settings and the factors associated with activation. This study examined patient portal activation among FQHC patients and assessed correlations with demographic, clinical, and health service use variables. We analyzed electronic health record data from adults >18 years old with at least one appointment. Data were accessed from the electronic health records for patients seen between 1 September 2018 and 31 August 2022 (n = 40,852 patients). We used multivariate logistic regression models to examine the correlates of having an activated EPIC-supported MyChart patient portal account. One-third of patients had an activated MyChart portal account. Overall, 35% of patients with an activated account had read at least one portal message, 69% used the portal to schedule an appointment, and 90% viewed lab results. Demographic and clinical factors associated with activation included younger age, female sex, white race, English language, being partnered, privately insured, non-smoking, and diagnosed with a chronic disease. More frequent healthcare visits were also associated with an activated account. Whether or not a patient had an email address in the EHR yielded the strongest association with patient portal activation. Overall, 39% of patients did not have an email address; only 2% of those patients had activated their accounts, compared to 54% of those with an email address. Patient portal activation rates were modest and associated with demographic, clinical, and healthcare utilization factors. Patient portal usage to manage one’s healthcare needs is increasing nationally. As such, FQHC clinics should enhance efforts to improve the uptake and usage of patient portals, including educational campaigns and eliminating email requirements for portal activation, to reinforce cancer prevention efforts.

## 1. Introduction

Federally Qualified Health Centers (FQHC) are safety-net centers that provide care to uninsured individuals or those who receive government insurance such as Medicare or Medicaid [1]. The population served by FQHCs is at greater risk for many cancers and suffers worse outcomes when they do develop cancer [2]. Although many efforts are being implemented in FQHC settings to improve cancer screening and prevention [3], studies have shown that they need further assistance in planning and implementing interventions that align with this goal, such as tobacco cessation treatment, screening for breast, cervical, and colorectal cancers, and reducing lifestyle-related risk factors (overweight/obesity and lack of physical activity).

Increasing uptake of electronic health record (EHR)-linked patient portals in FQHC settings could help transform prevention and control strategies for cancer and other chronic diseases [4]. Patient portals are communication tools that allow patients to access specific services by emphasizing provider continuity, planned and coordinated care, and self-management support [5]. They can also provide the centers with registries that could help them track patient populations, identify appropriate preventative services for patients, and notify providers when a patient is due to receive services if connected to the EHRs [6]. Further, research on patient portal use demonstrates downstream benefits for patients, including improved quality of care, enhanced knowledge, self-efficacy, and decision making [7], better medication adherence [8], increased use of preventative services [9], better symptom management [10], and better communication outcomes [11]. Patient portals also decrease unnecessary office visits and medical expenses but, most importantly, can improve cancer care health equity [12,13].

Approximately 90% of US healthcare institutions provide patients with access to their health information via the patient portal [14]. Despite widespread availability and demonstrated benefits, patient portal uptake has been limited [15]. Hong and colleagues examined portal use using the Health Information National Trends Survey (2020) data. They found that less than one-third of the adults surveyed were enrolled in a patient portal. Additional research has shown that patient portal uptake and use vary based on patient demographic variables, including health conditions, age, gender, race, and income level [16]. Given the data available on end-users, low-income patients appear to have the worst uptake of patient portals; unfortunately, poor uptake among vulnerable patients may increase the existing healthcare quality chasm and exacerbate health disparities [17]. As efforts continue to improve patient portal use, ongoing monitoring of inequalities and a better understanding of the factors driving differences in access will be vital to ensure that all patients can take advantage of digital tools to help them better manage their health. Monitoring the adoption of patient portal usage is especially important for patients receiving care in FQHCs due to the increased risk for cancer and other chronic diseases and well-established barriers to the uptake of technology-based health innovations.

There is limited research on patient portal use in FQHC settings. To address this significant gap in the literature, this study aimed to describe patient portal activation rates among patients treated in an FQHC system, determine whether activation rates varied based on demographic, clinical, or healthcare utilization variables, and describe the patient portal features used by patients. This information has implications for the development of targeted interventions for improving patient portal enrollment among vulnerable populations of patients with cancer and other chronic diseases receiving care in FQHC and other safety-net healthcare settings.

## 2. Methodology

### 2.1. Study Design

This secondary data analysis study describes EPIC MyChart patient portal activation rates and correlates of activation among patients receiving care at an FQHC. EPIC’s MyCHART is an online patient portal that allows patients to securely access and manage their healthcare information. This system enhances the patient’s experience by providing a convenient, user-friendly platform for interacting with healthcare providers and accessing personal health information. Key functions for patients include access to health records, appointment and medication management, communication with providers, access to test results, and billing information. By offering these features, EPIC’s MyCHART aims to empower patients with more control over their health care, improve communication between patients and providers, and streamline many aspects of health care management. To answer the study question, deidentified electronic health records of patients 18 or older were extracted from an extensive FQHC network and prepared for analysis through consultation with our Center for Clinical and Translational Science (CCTS). The study was approved by the Institutional Review Board of the University of Illinois Chicago (IRB # 2020-1621). All study activities took place during 2023.

### 2.2. Study Sample

The Mile Square Health Center (MSHC) EHR system was used to identify patients for this study. MSHC is a large FQHC in Chicago, Illinois, affiliated with the University of Illinois Chicago. MSHC has six clinic locations that serve a broad array of patients, most of whom fall below the federal poverty line. The analytic sample used for this study included N = 40,852 patients. Eligibility criteria for the study included being 18 years and older and having at least one appointment between 1 September 2018 and 31 August 2022. This time frame was used because EPIC was first launched by the FQHC in 2018, and the healthcare system launched an extensive COVID-19 vaccination campaign in the winter of 2021. All patients interested in vaccination were required to sign up for MyChart. Examining patient portal enrollment rates that included individuals who enrolled during the vaccination campaign would have overinflated estimates of typical patient portal uptake in the FQHC system. As such, we stratified the sample to reflect those enrolled prior to 2021 and those enrolled after 2021.

### 2.3. Study Variables

The study variables are described below.

#### 2.3.1. Demographic Characteristics

The following information was extracted from the patient’s EHR: age, race, primary language, sex, relationship status, insurance type, months as a patient, having an email address on file, and smoking status. Demographic information in the FQHC is collected at the time of patient intake and is entered into the EHR system by staff.

#### 2.3.2. Clinical Characteristics

International Classification of Diseases-10 (ICD-10) codes were used to identify patients with asthma, chronic obstructive pulmonary disease, hypertension, diabetes, cardiovascular disease, peripheral vascular disease, repiratory failure, cancer, bronchiectasis, and HIV (yes/no). The presence of a chronic health condition was dichotomized as 1 = 1 or more and 0 = None.

#### 2.3.3. Healthcare Utilization

The EHR for the FQHC clinics dates back to January 2010 and was reviewed in May 2021. Healthcare utilization variables included duration as a patient, the number of visits over this period, and the type of clinical visit (urgent care versus non-urgent care).

#### 2.3.4. Patient Portal Activation Status

A patient’s MyChart activation status was determined from their EHR and coded as 1 = activated and 0 = not activated. Those without an activated account were further classified as pending (e.g., an activation code was generated in the EHR, but no account was activated), information missing (e.g., no evidence of the patient being offered an opportunity to activate their account); or other (e.g., either the patient could not establish an account or declined participation).

#### 2.3.5. MyChart Patient Portal Activation Method

Patients could use two methods to activate an account: electronic activation (i.e., a staff member sends the patient a unique sign-up link via text or email through the EHR system at the end of the clinic visit) and self-activation (i.e., the patient is provided with a unique activation code on a post-visit summary sheet with enrollment information).

#### 2.3.6. MyChart Patient Portal Usage

Variables from the EHR to determine a patient’s use of the portal included whether the patient had read a portal message, replied to a portal message, scheduled an appointment, or viewed laboratory results (all categorized as yes/no).

### 2.4. Statistical Analysis

We examined bivariate descriptive statistics such as frequencies and conditional percentages by MyChart account status; simple logistic regression was used to test bivariate associations for inclusion in the model. Control variables included time as a patient (in months), type of care received (including visit type, i.e., urgent vs. non-urgent care and the number of visits), and economic status (e.g., insurance type), as these factors have been identified previously to impact patient portal utilization [15,16]. The majority of patients seen at MSHC fall below the federal poverty line. However, income is not uniformly documented in the MSHC patient records; we used private insurance to explain the economic heterogeneity. A multivariable logistic regression model was used to identify correlates of MyChart activation status. Among MyChart account holders, we summarized features used (e.g., reviewed lab results) within MyChart, stratified by those activated before January 2021 and those after, when efforts to engage patients to use MyChart were increased to prepare for COVID-19 vaccinations. Missing variable information was minimal and limited to biological sex (n = 14), race/ethnicity (n = 1522), language spoken (n = 943), insurance status (n = 129), and smoking status (n = 3285). The two variables with the highest percentage of missing data were race/ethnicity and smoking status. Smoking status, representing less than one percent of missing data (0.08), is a patient-level health behavior inconsistently documented in patient EHRs. Missingness for race/ethnicity (0.03%) was also minimal. Given the relatively low percentage of missing data and that a main effect was observed for race/ethnicity and smoking status in multivariate analyses, we do not believe that the complete case analytic approach introduced systematic bias.

## 3. Results

Table 1 displays patient MyChart Enrollment (N = 40,852). Among these, approximately one-third (34%) of patients had an activated MyChart account, with 58% of those users having activated their account before January 2021 (the start of the MyChart enrollment campaign at the FQHC to schedule COVID-19 vaccinations).

Of those without an activated MyChart account, 43% had a pending status (activation opportunity offered but not completed by the patient), 23% had no information in their EHR about an activation opportunity, and 34% were categorized as other (i.e., the code expired, the account was inactivated, possibly due to login failure, or the patient declined). Activation methods included electronic activation and self-activation using an access code. Electronic activation, in which a staff member sends a patient a unique sign-up link, was used for 35% of activated patients in our sample, and self-activation, in which a patient enters a patient-specific code and at least one other identifier in the sign-up website, was used for 45% of patients. Instant activation with assistance provided by a staff member shows a slightly higher activation success rate (50.6%) than when patients were required to log in independently (47.3%).

Bivariate analyses suggested associations between demographic, clinical, and healthcare utilization variables and MyChart activation (see Table 2). In bivariate analyses, demographic factors associated with having an activated MyChart patient portal account included age (30–44 years, 38.4%), female sex (40.0%), white race/ethnicity (40.5%), English language (37.2%), being in a relationship (37.9%), private insurance (44.3%), and having an email address on file (53.5%). Clinical variables associated with having a MyChart account included being a non-smoker (38.9%) and diagnosis of one or more chronic diseases (41.6%). Finally, more patients with frequent health care visits (>2) were more likely to have a MyChart account (39.7%). All bivariate relations with the MyChart account were significant at *p* < 0.001. As such, all variables were included in multivariate analyses.

Table 3 displays the results from the multivariable logistic regression model examining correlates of patient portal activation. The analyses included 85% of the sample (N = 34,724) with complete data after listwise deletion. Thirty-seven percent of patients included in the final multivariate analyses had an active MyChart account. The results of the multivariate models were similar to bivariate analyses with age, sex, race/ethnicity, language, relationship status, insurance type, smoking status, chronic conditions, and healthcare utilization significantly associated with portal activation. The adjusted rates estimated from the model are similar to the descriptive statistics in Table 2: 52% with an email address have a MyChart account compared to 3% without an email address (adjusted OR = 35.5, 95% confidence interval [CI] 31.5–39.9), after adjustment for all other variables in the model. We also considered the same model for predicting an activated MyChart account, excluding those without apparent access. The rate of MyChart activation was higher (46%) for this subgroup, yet models were very similar, with individual parameters differing by less than 6% and consistent statistical conclusions. While email address was still a significant predictor (OR = 18.2, 95% confidence interval 16.1–20.6), the magnitude of the effect was decreased, indicating having an email address was more important when including those not given an opportunity to access MyChart.

Finally, we examined the activation method and use of MyChart functions among users who activated their accounts before and after January 2021 (Table 4, N = 8165). Those who activated after 2021 were likelier to have set up their account using an activation code consistent with the email campaigns to increase access to MyChart and COVID-19 vaccines. Overall, 35% of patients with an activated MyChart account had read and replied to a message via the patient portal. Nearly 70% of patients (69.1%) had scheduled appointments, and 90.4% had viewed their lab results using MyChart. Those individuals with more recently activated MyChart accounts were less likely to have used all functions.

## 4. Discussion

Patient portals are a cost-effective strategy to improve patient access to health information and engagement in their care [7,8], particularly for individuals with cancer and other chronic health conditions. Most healthcare systems offer access to an EHR-linked patient portal. However, few studies have examined patient portal usage among individuals receiving care in an FQHC setting. This limited research on patient portal use among individuals served by safety-net centers represents a significant barrier to reducing disparities in cancer and other health outcomes. A key factor enabling patient portal enrollment is efforts by the healthcare system to provide access to patients. The results from our study suggest that two-thirds of all patients in our FQHC system had an opportunity to enroll in the patient portal (MyChart). However, only one-third of patients had enrolled in the portal. This enrollment rate is lower than the 40% reported by participants in a nationally representative sample of U.S. adults [18]), suggesting barriers to enrollment among patients in safety-net health settings. Portal activation rates were higher (46%) when we limited the sample to those with a documented opportunity to activate their MyChart account. There are different ways for patients to enroll in the MyChart system, and findings suggest that staff-assisted activation resulted in a higher activation rate than requiring patients to log in independently. These findings further support the importance of additional staffing at FQHC centers to support patients with patient portal activation processes [19].

Our multivariate analyses revealed that patient-level demographic, clinical, and healthcare utilization variables were associated with portal activation. Demographic factors related to MyChart activation. Although access to the Internet has improved with the widespread availability of smartphones, older adults with lower incomes are more reliant on government-issued phones or less expensive commercial models with limited Internet capabilities. Further, research suggests that older adults are more likely to have concerns about privacy and experience technological barriers to use [19]. Other demographic factors associated with lower uptake included male patients, patients of color, non-English language speakers, single patients, uninsured patients, and patients with public insurance. Many of these demographic variables are reported as barriers to healthcare access, including the uptake of available healthcare innovations in treatment and service provision [20,21]. These results suggest the substantial risk of continuing health disparities among underserved populations related to barriers to healthcare access, which is increasingly linked to technology-based care.

Clinical factors and healthcare utilization variables were also associated with patient portal enrollment. Diagnosis with a chronic illness was associated with portal activation. Our findings are consistent with research reporting an association between a chronic disease diagnosis and patient portal usage [22,23]. Patients with a chronic illness may receive more encouragement from providers to use the portal to monitor laboratory values, communicate about treatment, and access health promotion information. Healthcare utilization was also independently associated with MyChart activation, with patients with frequent clinic visits more likely to have an activated MyChart account. This finding is likely linked to the fact that patients with a chronic illness have more medical visits than those not experiencing a chronic disease. Additionally, higher activation rates among patients with more medical visits may be due to additional opportunities to learn about and receive help accessing MyChart or a perceived need for the resource to help manage their health care.

Our study contributes to the growing literature on demographic, clinical, and healthcare utilization variables associated with patient portal engagement. However, a new finding from this study points to the role of email addresses as a specific institutional-level barrier to patient portal access for lower-income patients. In our research, the most significant driver of MyChart patient portal enrollment status was whether a patient had an email address on file in their EHR. A substantial proportion (39%) of patients did not have an email address in their EHR. Of those without an email address, only 2% had a MyChart account, compared to 54% with an email address. Much has been written about the digital divide based on education, income level, and computer literacy skills [24]. As Internet access has increased over time, studies have suggested that inequalities in skill levels are more extensive than inequalities in access to the Internet [25]. For example, individuals with higher income levels generally have greater access to email and use it more frequently than those with lower income levels. This is attributed to better access to technology, internet services, and digital literacy among higher-income groups [26]. As such, a careful review of patient portal activation procedures and requirements (e.g., requiring an email address for patient enrollment) should be conducted to reduce institutional barriers to access. The University of Illinois Hospital and Health Sciences System requires an email address as a patient identifier and a date of birth for patients attempting to sign up online with an activation code. If a patient does not have an email address on file, they may not use the activation code provided on their after-visit summary. Moreover, having an email address on file is also required to change one’s MyChart password. Patients with no email address must rely on staff to enroll them through instant activation and reset their password through the EHR system if needed. Further efforts are required to establish activation processes that do not systematically exclude lower-income patients.

### 4.1. Implications of the Study Findings

Federally qualified health centers and other safety-net health centers provide critically important care for persons from lower socioeconomic backgrounds. Previously reported research has identified income as a barrier to the uptake of healthcare innovations such as patient portals [27]. The use of patient portals in the general population and our sample of FQHC patients remains low, presenting a significant challenge to the meaningful use of EHR to improve the management of chronic health conditions, including cancer. Indeed, sixty-five percent of MSHC patients were not enrolled in the patient portal despite most patients being provided with an opportunity to activate an account. These findings indicate a need to develop strategies to improve awareness of patient portals’ benefits and reduce barriers to enrollment. Systematic reviews have examined patient-perceived barriers and facilitators to patient portal use [28,29]. Facilitators to enrollment included provider encouragement, the desire to access their health information, and the opportunity for improved communication with their providers. Barriers to uptake were captured in two themes: lack of awareness of the patient portal and data safety and security concerns. Themes related to these two barriers and the desire to receive direct assistance in enrolling in the patient portal were also identified in a qualitative study by Matthews and colleagues [19].

Clinic-based strategies have implications for increasing trust in the patient portal and the reduction of other specific barriers. For example, providers and staff play an instrumental role in recommending and providing resources for portal enrollment. Based on findings from the current study and the extant literature, a range of patient-centered approaches should be considered to increase the uptake of patient portals. First, due to the literature supporting limited awareness of the patient portal among FQHC patients [19], we recommend increases in clinic-wide outreach and educational campaigns. Clinic-based patient portal enrollment campaigns should prioritize several vital issues. The first is to develop messages emphasizing data safety and security and highlighting the system’s features most appeal to patients, including scheduling appointments, contacting their providers, and obtaining test results [19]. Focusing on these features may help to increase the perceived relevance of the portal for individuals [30]. In addition, the information provided during the enrollment campaign should be developed with attention to low health literacy and computer literacy skills. Educational materials about patient portal enrollment and usage can be distributed to patients via mailings or in the clinic environment. As part of a larger educational campaign, providers and staff should be trained to promote patient portal use and integrate portal enrollment into routine care and workflows for all patients. For example, beginning in September 2020, MSHC now includes information about MyChart activation in all after-visit summaries provided to patients. Although a novel approach, the extant literature and the results of this study suggest that the lower levels of computer literacy among some FQHC patients may necessitate more staff support to facilitate portal enrollment.

Patient navigators represent a practical and cost-effective strategy to help patients overcome barriers to health care services [31]. Patient navigators can be used in clinical settings to provide patients with educational information about patient portals, assist with activating accounts using computers or smaller hand-held tablets, and answer questions about navigating the patient portal environment. Further, patient navigators can give patients the option of receiving a follow-up call within the next 3–5 business days to help patients problem-solve difficulties they may encounter when trying to access the site independently. These combined activities provided by patient navigators may help overcome barriers to use based on computer literacy skills.

Matthews and colleagues conducted a recent pilot study [32] to examine the benefits of using patient navigators to assist patients in an FQHC setting (MSHC) in enrolling in a patient portal. Four trained patient navigators offered activation assistance to 83% of eligible patients, with 64% (n = 1062) accepting help. Among those who accepted assistance, 74% of patients without prior MyChart enrollment activated their accounts during the clinic visit. Patient portal activation rates increased from 44% in the eight months before navigators were present to 51% during the eight months with navigators. Significant increases in portal usage were observed, particularly for viewing lab results and reading messages, with notable increases among African American (44% to 49%) and Latinx patients (52% to 60%). The study indicates that employing patient navigators is a viable and effective strategy for boosting enrollment in Federally Qualified Health Centers patient portals. However, future research must develop implementation strategies for adapting patient navigation strategies for local context-related patient-, clinic-, and system-level barriers.

### 4.2. Limitations

Study limitations should be noted. First, this is a single-center observational study. As such, the findings may not generalize to other FQHC locations. Our definition of an active portal user was based on previous research, i.e., a patient who used the portal ≥1 time within 12 months after account activation [28]. However, there is no gold standard for defining active portal users. It is unclear if these findings generalize to other FQHC systems. We did not have data about barriers to portal use among patients who were provided an opportunity to activate their MyChart account but did not. However, qualitative findings from patients from the same FQHC [19] suggest the essential roles of a lack of awareness of the portal and relevant functionality, a stated preference for talking directly to the provider, and a lack of computer literacy, to name a few. However, additional research is needed to examine further barriers to enrollment among patients who were given access to portal enrollment but who did not.

## 5. Conclusions

The management of chronic diseases, including cancer, remains a significant public health priority. Emerging evidence suggests that patient portals improve the quality of patient care and reduce health inequalities [4]. While the uptake of patient portals has increased, the promotion and use among FQHC patient populations remain low, which has consequences for cancer and other chronic disease management among patients receiving care in safety-net health centers. Study findings highlight the relationship between patient-level demographic, clinical, and healthcare utilization factors on portal enrollment rates. Institutional factors, such as portal enrollment procedures that disadvantage lower-income patients, such as the necessity of email addresses, were newly identified but modifiable barriers to enrollment. Institutional-level factors are critical social determinants of health. Further research requires the identification and removal of similar institutional barriers to care. Additionally, innovative approaches to increase patient portal enrollment efforts in FQHC settings are needed, such as educational campaigns targeting low-health literacy populations and using a patient navigator or other support staff to provide direct assistance with patient portal enrollment.

## Figures and Tables

**Table 1 healthcare-12-01505-t001:** Percentage of FQHC patients enrolled in MyChart Patient Portal, July 2021.

	MyChart Enrollment (N = 40,852)			
	Not Enrolled		Enrolled	
	No Information	Not Activated	Before January 2021	After January 2021
Number	9435	17,435	8165	5817
Percentage	23.10%	42.70%	20.00%	14.20%
Total n, %	26,870 (65.8%)		13,982 (34.2%)	

**Table 2 healthcare-12-01505-t002:** Demographic characteristics of Patients Enrolled in MyChart (N = 40,852).

Enrolled in MyChart *						
	No		Yes		Total	
	N	Row %	N	Row %	N	Column %
Number of Health Care Visits						
More than 2	15,045	60.3	9923	39.7	24,968	61.1
1 or 2 visits	11,825	74.4	4059	25.6	15,884	38.9
Total	26,870	65.8	13,982	34.2	40,852	100
Appointment Type						
Primary Care	22,637	65.1	12,161	34.9	34,798	85.2
Urgent Care	4233	69.9	1821	30.1	6054	14.8
Total	26,870	65.8	13,982	34.2	40,852	100
Age category						
18–29	6927	65.9	3589	34.1	10,516	25.7
30–44	8664	61.6	5403	38.4	14,067	34.4
45–64	8321	67.4	4021	32.6	12,342	30.2
65+	2958	75.3	969	24.7	3927	9.6
Total	26,870	65.8	13,982	34.2	40,852	100
Biological Sex						
Female	16,362	60	10,906	40.0	27,268	66.8
Male	10,495	77.3	3075	22.7	13,570	33.2
Total	26,857	65.8	13,981	34.2	40,838	100
Race/Ethnicity						
White	2103	59.5	1432	40.5	3535	9
Black	13,817	64.6	7565	35.4	21,382	54.4
Hispanic	8336	70.7	3453	29.3	11,789	30
Other	1634	62.3	990	37.7	2624	6.7
Total	25,890	65.8	13,440	34.2	39,330	100
Language Spoken						
English	22,234	62.8	13,192	37.2	35,426	88.8
Spanish	3408	83.8	658	16.2	4066	10.2
Other Language	320	76.7	97	23.3	417	1
Total	25,962	65.1	13,947	34.9	39,909	100
Email Address on file						
No	15,030	97.8	340	2.2	15,370	37.6
Yes	11,840	46.5	13,642	53.5	25,482	62.4
Total	26,870	65.8	13,982	34.2	40,852	100
Partner/marital status						
Not Partnered	20,525	65.8	10,679	34.2	31,204	79.7
Partnered	4926	62.1	3003	37.9	7929	20.3
Total	25,451	65	13,682	35	39,133	100
Insurance type						
Public or no insurance	18,724	71.2	7574	28.8	26,298	64.6
Private	8029	55.7	6396	44.3	14,425	35.4
Total	26,753	65.7	13,970	34.3	40,723	100
Current smoker						
No	18,357	61.1	11,696	38.9	30,053	80.0
Yes	5604	74.6	1910	25.4	7514	20.0
Total	23,961	63.8	13,606	36.2	37,567	100
Diagnosed with a Chronic condition						
None	16,121	71.8	6335	28.2	22,456	55.0
1 or more	10,749	58.4	7647	41.6	18,396	45.0
Total	26,870	65.8	13,982	34.2	40,852	100

* All bivariate relations with the MyChart account were significant, *p* < 0.001. Some variables had missing data in the electronic medical record.

**Table 3 healthcare-12-01505-t003:** Results of Multivariate Logistic Regression Models Examining Factors Associations with Patient Portal Activation (N = 34,687).

N = 34,687	Odds Ratio	Std. Err.	z	*p* > z	[95% Conf. Interval]	
Age category						
18–29	1	(base)				
30–44	1.03	0.04	0.85	0.395	0.96	1.10
45–64	0.88	0.04	−3.26	0.001	0.81	0.95
65+	0.82	0.05	−3.14	0.002	0.73	0.93
Biological Sex						
Female	1	(base)				
Male	0.66	0.02	−13.37	0	0.62	0.70
Race/Ethnicity						
White NH	1	(base)				
Black NH	0.60	0.03	−10.13	0	0.55	0.67
Hispanic	0.70	0.04	−6.54	0	0.63	0.78
Other	0.86	0.06	−2.22	0.027	0.75	0.98
Language						
English	1	(base)				
Spanish	0.72	0.05	−5.08	0	0.63	0.82
Other Language	0.76	0.12	−1.72	0.085	0.56	1.04
Partner/marital status						
Not Partnered	1	(base)				
Partnered	1.23	0.05	5.63	0	1.15	1.33
Insurance type						
Public or no insurance	1	(base)				
Private	1.44	0.04	12.1	0	1.36	1.52
Smoking status						
Everyday/Heavy	1	(base)				
Somedays/Light	1.19	0.08	2.54	0.011	1.04	1.36
Former	1.37	0.07	5.78	0	1.23	1.52
Never	1.46	0.07	8.29	0	1.33	1.59
Chronic conditions						
None	1	(base)				
1 or more	1.52	0.05	13.35	0	1.43	1.61
Number of Health Care Visits						
More than 2	1	(base)				
1 or 2 visits	0.82	0.03	−5.92	0	0.76	0.87
Number of Years as a Patient						
Duration (years)	1.10	0.00	22.47	0	1.09	1.11
Type of Appointment						
Primary Care	1	(base)				
Urgent Care	1.15	0.05	3.12	0.002	1.05	1.26
Email Address on File						
No	1	(base)				
Yes	35.47	2.14	59.13	0	31.52	39.93
Intercept	0.02	0.00	−44.07	0	0.02	0.02

**Table 4 healthcare-12-01505-t004:** Comparison of MyChart Activation Before and After COVID-19 Vaccination Campaign.

Activation Date *						
	Before 2021		After 2021		Total	
	N	%	N	%	N	%
MyChart Activation Method						
Electronic activation	4787	58.6	1915	32.9	6702	47.9
Patient Activation	3375	41.4	3902	67.1	7277	52.1
Total	8162	100	5817	100	13,979	100
Message Read						
No	4449	54.5	4637	79.7	9086	65.0
Yes	3716	45.5	1180	20.3	4896	35.0
Total	8165	100	5817	100	13,982	100
Replied to Message						
No	4449	54.5	4637	79.7	9086	65.0
Yes	3716	45.5	1180	20.3	4896	35.0
Total	8165	100	5817	100	13,982	100
Scheduled Appointment						
No	1846	22.6	2478	42.6	4324	30.9
Yes	6319	77.4	3339	57.4	9658	69.1
Total	8165	100	5817	100	13,982	100
Viewed Lab Results						
No	541	6.6	807	13.9	1348	9.6
Yes	7624	93.4	5010	86.1	12,634	90.4
Total	8165	100	5817	100	13,982	100

* All bivariate relations with activation after January 2021 were significant, *p* < 0.001.

## Data Availability

Data are contained within the article.

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
