# Peer review of "Factors Associated with Uptake of Patient Portals at a Federally Qualified Health Care Center"

_healthcare, 2024, doi:10.3390/healthcare12151505_

Round 1

Reviewer 1 Report

Comments and Suggestions for Authors

About the title:

What type of factors (demographic, clinical and use of health services)? I suggest adding them to the title and keywords.

Use of WEB portals? For cancer patients?

The study began in 2018 and will be published 6 years later (2024). How the authors justify that the study is valid and that its results are still valid.

It is not clear why the authors chose cancer for this study. Does it apply to all types of cancer?

They used: International Classification of Diseases-10 (ICD-10) codes were used to identify patients with asthma, chronic obstructive pulmonary disease, hypertension, diabetes, cardiovascular disease, peripheral vascular disease, respiratory failure, lung cancer, bronchiectasis, and HIV. Does that mean it only applies to lung cancer?

If patient portal activation rates were modest and associated with demographic, clinical, and healthcare utilization factors, what actions have been taken to increase its impact: Is there information from educational campaigns?

What was the approval and protocol of the Ethics Committee approved?

The study was approved by the Institutional Review Board of the University of Illinois Chicago (IRB # 87 2020-1621). But, did the study start in 2018?

Table 2, 3 and 4 are confusing to understand. Several rows without data. The authors must change their design. And perhaps it is better to show analysis of the data.

The Discussion section should focus on the differences with respect to the factors: demographic, clinical and use of health services, separately

The current conclusions are very few and lacking forcefulness.

It is necessary for the authors to update the references, several are more than 10 years old and have already lost scientific validity.

Comments on the Quality of English Language

The grammar used is confusing to understand. The texts must be written in a technical and scientific way. The explanation of the tables is very confusing due to the wrong use of syntax

Author Response

Thank you for your feedback on our manuscript.  We have addressed each of your comments.  Revisions to the document are highlighted in yellow. 

    1. Clarity of the dates of the research study.

    Response:  Secondary data analyses were conducted as part of a study approved in 2020 and completed in 2021.  The EHR-derived patient sample was restricted to all patients seen between 2018 and 2020. 

    1. Does the study include all cancers?

    Response:  This study was not limited to cancer patients.  Instead, the study aimed to examine the update of the patient portal among high-risk patient populations served in an FQHC setting.  This information has important implications for cancer patients as persons from lower socio-economic backgrounds are at elevated risk for cancer.  Knowledge of portal enrollment rates and associated factors has implications for cancer prevention and control initiatives offered for this vulnerable patient population.

    1. When was IRB approval obtained?

    Response:  The study was approved by the Institutional Review Board of the University of Illinois Chicago (IRB # 2020-1621). All study activities took place during 2021.  

    1. When did the study start?

    Response: Secondary data analyses were conducted as part of a study conducted in 2021.  The patient sample was limited to 2018-2020.  This time frame was selected because the FQHC center launched EPIC and MyChart in 2018.  The data sample was limited to 2020 due to the launch of a COVID-19-related vaccination protocol. All patients scheduling vaccination appointments had to enroll in the patient portal to facilitate surveillance and vaccination monitoring. Individuals who were not regular patients at the FQHC were given opportunities to receive a vaccination at the clinic locations.  The inclusion of rates during this period would have resulted in a patient sample not representative of the typical patient population.

    1. Revise Tables 1-3

    Response:  All tables were edited to remove the additional lines and empty spaces. 

    1. Revise the discussion section to describe the complete findings.

    Response:  The discussion section was revised as requested.

    1. Revise the conclusions section to increase impact.

    Response: The discussion section was revised as requested. 

    1. Edit the manuscript for grammatical errors

    Response:  The manuscript was edited as requested. 

Reviewer 2 Report

Comments and Suggestions for Authors

1. Since many of the features in EPIC are tailored to the implementing centre, it would be helpful for the authors to include information about what is actually available to patients when they log into the patient portal at their centre (are they getting PROs, tailored advice, just viewing test results and upcoming appointments, is messaging turned on, are there education materials, etc). Also, how are they invited to join the patient portal?  

2.  There were a lot of sociodemographic variables collected- where/ who enters this (is this provided by the patient, entered by a staff member on registration?); also how much missingness was there?

3. Since relatively few patients had a email on file, it is not clear how registration for MyChart was offered to patients and how this was documented.

4. The formatting of Table 2 makes it difficult to follow.

5. It is not clear why the authors chose to use complete case analysis for the regressions; how does this potentially bias the results?

6. The formatting of Table 3 makes it really hard to follow.

7.  Did you look at potential co-linearity amongst variables that you included in the multivariable regression models? It also isn't entirely clear what the primary exposure is for the multi-variable regression.

8. Table 3 indicates multivariate modelling, but you have described multivariable logistic regression (these are two different things).  Please correct.

9. Why did the authors not evaluate sociodemographic and clinical characteristics associated with patient activation of the portal rather than any activation?  Is electronic activation not done by the cancer center vs patient activation done by the patient?  These would mean two separate things- if there are sociodemographic differences in who has a portal that the cancer centre activated then there are issues with equity delivery of care/ support; if there are sociodemographic differences in patient activations then this tells you who/ what you might need to target to improve uptake/ accessibility.

10.  The discussion does not match the findings.  The types of facilitators and potential strategies to improve uptake do not match the barriers that you have described.  ie:  if there is poor uptake in older adults, what specifically would you do to fix that?  Quite a bit of space is dedicated to literacy/ digital literacy which you did not measure/ evaluate.

11. It is not clear to me from the discussion what this study adds to the existing body of literature, especially without an accompanying explanatory qualitative study understanding the underlying, context-specific barriers leading to these findings.

Author Response

Response to Reviewers

Thank you for your feedback on our manuscript.  We have addressed each of your comments.  Revisions to the document are highlighted in yellow. 

  1. Include information about how patients enrolled in the MSHC FQHC patient portal.

Response:  This information has been added to the document.  

  1. Add details regarding missing data on demographic and clinical variables.

Response: This information is included in the document.  

  1. How was registration offered to patients without an email address?

Response: This information is included in the document.  

  1. It is unclear why the authors used complete response data.

Response:  Missing data were minimal, except non-critical variables (i.e., smoking status).  This information is included in the document.

  1. The formatting of Table 2 makes it difficult to follow

Response:  Tables 1-3 were reformatted for ease of reading.

  1. Collinearity of variables included in the model

Response:  As one might expect, there was a correlation between demographic variables, including race/ethnicity and insurance type.  However, multiple regression aims to determine, among multiple correlated explanatory variables, the direct effects of each on the response. Correlations among predictor variables are not a problem for numerous regression; instead, they are part of the information that multiple regression uses to infer direct effects, as opposed to overall relationships. 

  • Morrissey, Michael B. & Ruxton, Graeme D. (2018). Multiple Regression Is Not Multiple Regressions: The Meaning of Multiple Regression and the Non-Problem of Collinearity. Philosophy, Theory, and Practice in Biology 10 (3).

  1. What was the primary exposure for the multivariable regression models

Response:  The primary dependent variable was an Activated MyChart Account. Independent variables include demographic, clinical, and healthcare utilization variables. 

  1. Correct labeling of Table 3

Response: The description of Table 3 was revised

  1. Examination of the role of clinical and demographic variables on patient portal activation

Response: These variables were examined in the models described.  See pages xx.

  1. The clinical setting

Response: The clinical setting was an FQHC primary care setting.  The FQHC was not connected to a cancer center.

  1. Revise the discussion:

Response: The discussion was edited as suggested. 

Round 2

Reviewer 1 Report

Comments and Suggestions for Authors

The authors of this manuscript have reviewed and corrected all comments and observations made by the reviewer. In my opinion it can be published.

Author Response

Thank you for your efforts in reviewing our manuscript.  I understand that you don't require any additional changes at this point.  

Reviewer 2 Report

Comments and Suggestions for Authors

Thank you for addressing the previous comments, the changes have improved the manuscript.

1. Introduction: in the added/ modified text, "reducing" should be corrected to "increasing/ improving"

2. Methods/ Results: it is still not clear how the variables for the multivariable regression models were selected, was this ad hoc, forward selection etc?

3. Results: is there any potential bias introduced by using complete case analysis?  Is there a systematic reason why certain variables were missing for certain patients?

4. Discussion: "There are different ways for patients to enroll in the MyChart system, and findings suggest that staff-assisted activation resulted in a higher activation rate than requiring patients to log in independently. These findings indicate the importance of additional staffing at FQHC centers to support patients with patient portal activation processes." This statement is supported by the literature, it would be best to add a reference.

5. Discussion: "However, a new finding from this study points to the role of institutional factors as a barrier to patient portal access for lower-income patients." This is not a novel finding; the vast majority of articles examining penetration/reach/access/adoption of digital health technologies cite a "digital divide" and structural barriers/ determinants of health for lower-income patients are well documented. You even have a citation in the next paragraph saying that this has been reported before.

6. Discussion: ". However, future research must address various patient-, clinic- and system-level barriers".  Navigation is a way to address barriers; this statement would make more sense if you indicated that work would be needed to adapt it to local contexts.

Author Response

  1. Introduction: in the added/ modified text, "reducing" should be corrected to "increasing/ improving"

Revised as suggested

  1. Methods/ Results: it is still not clear how the variables for the multivariable regression models were selected, was this ad hoc, forward selection etc?

See response in text

  1. Results: is there any potential bias introduced by using complete case analysis?  Is there a systematic reason why certain variables were missing for certain patients?

See response in text

  1. Discussion: "There are different ways for patients to enroll in the MyChart system, and findings suggest that staff-assisted activation resulted in a higher activation rate than requiring patients to log in independently. These findings indicate the importance of additional staffing at FQHC centers to support patients with patient portal activation processes." This statement is supported by the literature, it would be best to add a reference.

Added

  1. Discussion: "However, a new finding from this study points to the role of institutional factors as a barrier to patient portal access for lower-income patients." This is not a novel finding; the vast majority of articles examining penetration/reach/access/adoption of digital health technologies cite a "digital divide" and structural barriers/ determinants of health for lower-income patients are well documented. You even have a citation in the next paragraph saying that this has been reported before.

See response in text

  1. Discussion: ". However, future research must address various patient-, clinic- and system-level barriers".  Navigation is a way to address barriers; this statement would make more sense if you indicated that work would be needed to adapt it to local contexts.

See response in text